# Sealing Ability of Bioactive Root-End Filling Materials in Retro Cavities Prepared with Er,Cr:YSGG Laser and Ultrasonic Techniques

**DOI:** 10.3390/bioengineering9070314

**Published:** 2022-07-14

**Authors:** Ivona Bago, Ružica Lucić, Ana Budimir, Valentina Rajić, Merima Balić, Ivica Anić

**Affiliations:** 1Department of Endodontics and Restorative Dentistry, School of Dental Medicine, University of Zagreb, Gunduliceva 5, 10000 Zagreb, Croatia; vbrzovic.rajic@sfzg.hr (V.R.); anic@sfzg.hr (I.A.); 2Health Centre Orašje, Orašje, Bosnia and Herzegovina, 76270 Orasje, Bosnia and Herzegovina; rlucic1991@gmail.com; 3Department of Clinical and Molecular Microbiology, Clinical Hospital Centre Zagreb, Kispaticeva 12, 10000 Zagreb, Croatia; abudimir@kbc-zagreb.hr; 4School of Dental Medicine, University of Zagreb, 10000 Zagreb, Croatia; merimica.balic@gmail.com

**Keywords:** root-end filling, mineral trioxide aggregate, ultrasonic, laser, microleakage, bioceramics

## Abstract

The aim of this in vitro study was to compare the apical sealing ability of total fill bioceramic root repair material (BC-RRM) and mineral trioxide aggregate (MTA), regarding the retrograde preparation technique used: ultrasonic or erbium, chromium: yttrium, scandium, gallium, or garnet (Er,Cr:YSGG) laser. The study sample consisted of 48 human single-rooted teeth. After root-end resection, the samples were divided into two groups, according to the retrograde preparation technique used: Group 1: ultrasonic; Group 2: Er,Cr:YSGG laser. In each group, half of the retrograde cavities were filled with BC-RRM, and the other half were filled with MTA. The specimens were mounted in tubes and sterilized in plasma. The root canals were inoculated with *Enterococcus faecalis*, and the tubes were filled with fetal bovine serum, leaving the apical part of the root in the serum. After 30 days, the canals were sampled and cultured, and the colony forming units (CFUs) were counted with the additional polymerase chain reaction (PCR analysis). There was no significant difference between ultrasonic groups and the Er,Cr:YSGG-MTA group, regarding the number of CFUs (*p* > 0.05). The Er,Cr:YSGG-BC-RRM group showed the highest number of remaining viable bacteria (*p* < 0.001). Both filling materials filled in ultrasonic preparations presented similar sealing abilities. The BC-RRM showed more leakage when used in retro cavities prepared with the Er,Cr:YSGG laser.

## 1. Introduction

Periapical surgery includes the removal of pathological periapical tissue, root-end resection, and root-end cavity preparation with the placement of a retro-filling material [1]. It is indicated when nonsurgical root canal treatment or retreatment is not possible or has been unsuccessful [2]. The aim of an apicoectomy is the removal of the apical third of the root, which is very often colonized with bacteria that can compromise the healing of periapical tissue [3]. The root-end filling material is placed in the retro-cavity, in order to prevent the leakage of bacteria and their byproducts in the periapical area and vice versa [4,5].

Modern techniques of root-end cavity preparation include ultrasonic devices and lasers [6,7]. The advantage of ultrasonic microsurgical tips is a better adaptation to root anatomy, thus conserving the root canal wall. Additionally, less exposure of dentinal tubules after ultrasonic preparation decreases the chances of leakage around the root-end filling material [8]. Compared to bur preparation, ultrasonic tip preparations are cleaner, smoother, and contain less debris and smear layers [9]. However, there have been concerns regarding the microfractures in dentinal walls after ultrasonic tip preparation [9,10]. With the use of an erbium laser, a lower incidence of microcracks was demonstrated as a result of non-contact work [11]. Additionally, erbium lasers have a superficial bactericidal effect and remove the smear layer, leaving a smooth and cleaned resected surface, which provides the better adaptation of retro-filling material in the retro-cavity [11,12,13]. Studies comparing the influence of ultrasonic and Er:YAG laser root-end cavity preparation on the microleakage of retro-filling materials yielded contradictory results, regardless of the material used [14,15,16].

MTA is the most commonly used root-end filling material, due to its moisture tolerance, sealing ability, marginal adaptation, bioactive characteristics [17,18], cementogenesis potential, and dentinogenic activity [18,19]. It is composed of Portland cement, with 4:1 addition of bismuth oxide. The cement is made up of calcium, silicon, and aluminium, with the main constituent phases of tricalcium and dicalcium silicate and tricalcium aluminate [20].

Previous studies reported on the good sealing ability and marginal adaptation of MTA over 12-week and 12-month periods, compared to other retro-filling materials [21,22]. Additionally, it has been shown that MTA in retrocavities prepared by erbium lasers leaked less than in ultrasonic retrocavities [6]. However, handling properties, long setting time, and low washout resistance during the initial phase of setting complicates its use for many practitioners [23].

An alternative to MTA are bioceramic-based materials, which set faster and have better handling properties. The newly developed total fill root repair material (RRM) (FKG Dentaire, La Chaux-de-Fonds, Switzerland) is a premixed material in putty form that is composed of calcium silicates, zirconium oxide, tantalum oxide, and calcium phosphate monobasic. Recent studies on this material showed high healing rates after its use in periradicular surgery [24], as well as biocompatibility and cytotoxicity comparable with MTA [25,26]. An animal study by Chen et al. [27] showed a better healing rate histologically after the placement of BC-RRM, compared to MTA. On the other side, studies comparing the sealing ability between BC-RRM and MTA yielded contradictory results [28,29].

The sealing ability of root canal sealers has been evaluated with different leakage tests, i.e., the dye leakage, bacterial leakage, fluid filtration model, leakage of fluorescent microspheres, glucose leakage tests [30,31]. Because microorganisms are the cause of apical periodontitis, bacterial leakage tests with viable bacteria as markers could be more clinically relevant for the comparison of the sealing ability of different endodontic materials [32].

The aim of this study was to compare the sealing ability of MTA and bioceramic root-end filling materials in root-end cavities prepared with the erbium, chromium: yttrium-scandium-galium garnet (Er,Cr:YSGG) laser, and ultrasonic techniques.

## 2. Materials and Methods

### 2.1. Sample Preparation

This study was approved by the local Ethical Committee No 05-PA-26-11/2015. The study sample consisted of 48 human extracted single-rooted mandibular and maxillary first and second premolars. All teeth were scanned in cone beam computed tomography (CBCT) (field of view, 5 mm × 5 mm; ENDO, 85 µm; 6.3 mA; 90 kV; 8.7 s; 450.3 mGycm2) (Cranex 3DX, Soredex, Tusula, Finland), in order to select only those with a single straight canal.

The selected teeth were cleaned and prepared according to the protocol described by Balic et al. [33]. Each tooth was stored in 0.5% chloramine-T solution at 4 °C after extraction. The working length (WL) was established at 14 mm by decoronation of each tooth using a water-cooled diamond fissure bur # 016 (Komet, Rock Hill, SC, USA). Teeth with wide root canals that could be easily passed with a # 20 K-file (Dentsply Maillefer, Ballaigues, Switzerland) without hatch were not included in the study. The root canals were instrumented with rotary ProTaper Next technique (Dentsply/Maillefer, Tulsa, OK, USA) at 300 rpm speed and torque 2.7 Nm, with instruments X1 (17/04), X2 (25/06), and X3 (30/07). During the instrumentation, the root canals were irrigated with 5 mL of 3% sodium hypochlorite solution (NaOCl) using a 30G needle and 5 mL syringe (BD, Microlance, Becton Dickinson, Madrid, Spain). After instrumentation, the canals were rinsed with 1 mL of 15% ethylenediaminetetraacetic acid (EDTA) (Calsinase, Lege Artis, Dettenhausen, Germany), which was left in the canal for 1 min, and then irrigated with 1 mL of 3% NaOCl and 1 mL of saline solution using a 30G needle and syringe. The root canals were dried with sterile paper points (ProTaper Next X3, Dentsply).

### 2.2. Root-End Cavity Preparation

The apical part of the prepared roots was resected for 3 mm perpendicular to the long axis of the sample using diamond fissure bur No 016 (Komet, Rock Hill, SC, USA) with water cooling.

The prepared samples were randomly distributed in two main experimental groups (n = 20 samples/each), according to the root-end cavity preparation technique used.

Group 1. (US): A stainless-steel ultrasonic preparation tip of the ultrasonic device (Piezon Master 400, EMS, Switzerland) was used to prepare the retro cavities at the root apex. The ultrasonic device was set at medium mode.

Group 2. (Er,Cr:YSGG): The Er,Cr:YSGG laser (wavelength: 2780 nm, Biolase, San Clemente, CA, USA) was set at the following parameters: power: 3.5 W, frequency: 25 Hz, water: 85%, air: 80%, and fluence: 28 J/cm^2^, and it was used to prepare the retro cavities. A MZ8 laser tip (diameter 800 μm) was used.

All cavities were prepared by the same operator. During preparations, the samples were moist. One operator did all the preparations during the standardized time of 15 s.

In the positive and negative control groups (n = 4/each), both the ultrasonic technique (two samples) and Er.Cr:YSGG laser (two samples) were used with the same protocols described before.

The samples were fixed in plastic 1 mL Eppendorf threaded tubes (Eppendorf, Hamburg, Germany) through a hole made in the cap of the tube with the apical part facing the bottom of the tube and the coronal part at the level of the cap. The samples were fixed with flowable composite resin (Universal Flo, GC, Tokyo, Japan), which was polymerized with the polymerization lamp (Bluephase, IvoclarVivadent, Schaan, Lichtenstein) for 40 s and cyanoacrylate (UHU, Bühl, Germany) at the cap/sample interface.

### 2.3. Root-End Filling Procedure

The samples in both groups (n = 20 samples per group) were divided into two subgroups (n = 10/each subgroup): *Subgroup 1*: root canal filled with a bioceramic material (BC-RRM TotalFill Root Repair Material, FKG Dentaire, La Chaux-de Fonds, Switzerland); *Subgroup 2*: root canals were filled with MTA (MTA Angelus, Londrina, Brasil).

Firstly, a gutta-percha point size X3 (ProTaper Next, Dentsply) with shortened 3 mm of the tip was placed in the root canal, in order to act as a matrix for retro-filling material. The BC-RRM and MTA were prepared and used according to the manufacturer’s instructions.

After the obturation, the outer surface of the root up to the resected area was covered with two layers of nail varnish.

The retro cavities of positive controls were filled with a temporary material (Cavit G, 3M ESPE, Neuss, Germany) and compressed with a cold condenser. In the negative controls, the retro cavities were filled in with pink wax (Cavex, Haarlem, Germany), and the whole apical surface and material were covered with two layers of nail varnish [30].

All samples were incubated at 37 °C in 100% humidity for two weeks. Afterwards, the samples were sterilized in plasma (PLASMA; Sterrad 100S, Johnson&Johnson, Irvine, CA, USA), and the control of sterilization control was performed in four additional samples using culture method. The microbiological samples were collected from four samples and prepared for the culture method. After the incubation period, the blood agar plates were checked for whether there was any bacterial growth. This procedure was used to confirm the sterilization of all sterilized samples

### 2.4. Bacterial Nutrient Leakage Model

All microbiological procedures were done inside a sterilized cabinet under aseptic conditions. Bacterial nutrient leakage model was conducted according to Antunes et al. [29].

The Eppendorf tubes were filled in with fetal bovine serum (Sigma-Aldrich, Taufkirchen, Germany), so that the apical part of the specimen was dipped in the serum.

In this study, *Enterococcus faecalis* ATCC 29212) bacterial suspension was used. It was prepared from pure culture of the bacteria with trypticase soy broth (TSB) to get the suspension of 0.5 McFarland (Densimat, BioMerieux, Marcy i’Etoile, France). The 5 µL of the prepared bacterial suspension was inoculated in each root canal of both experimental groups and control groups using a sterile insulin syringe and needle (0.3 × 8 mm), without overflowing. The samples were coronally closed with another sterile cap of the Eppendorf tube.

The prepared samples were incubated at 37 °C in 100% humidity for 30 days. After the incubation period, the microbial root canal samples were collected and prepared for the culture method and polymerase chain reaction (PCR). The sampling technique and dilution protocol was made according to Balic et al. [33]. After 10-fold serial dilutions of the initial sample, an aliquot of 10 µL from each diluted tube was plated on blood agar plates (211037, Becton Dickinson, NJ, USA) and incubated for 24 h at 100% humidity and 37 °C. After the incubation period, the number of colonies on plates was counted and transformed into actual colony forming units (CFU), based on the dilution factor.

### 2.5. Polymerase Chain Reaction

The polymerase chain reaction (PCR) was used to confirm the presence of *E. faecalis* and exclude the possibility of false-negative results The PCR analysis was conducted according to the already published protocol [34]. Conditions for PCR reaction were optimized by repeated reactions, and *E. faecalis* (ATCC 29212) was used as positive control. Six primers (each 0.5 µmol L^−1^) and two units of recombinant Taq DNA polymerase (Cinnagan Inc., Tehran, Iran) were used. The following primers, based on the whole *E. faecalis* V583, were used:

E16F (AGAGTTTGATCCTGGCTCA) and Ef16R (GGTTACCTTGTTACGACTTC); product 1522 bp;

EfisF (ATGCCGACATTGAAAGAAAAAATT) and EfisR (TCAATCTTTGGTTCCATCTCT); product 803 bp;

EfesF (GTGTTAAAACCATTAGGCGAT) and EfgsR (AAGCCTTCACGAACAATGG); product 650 bp.

Gel electrophoresis was performed on 1% agarose gel (Cinnagen, Tehran, Iran) for gel electrophoresis reaction (Akhtarian, Tehran, Iran). The molecules in the gel were stained with ethidium bromide (Merck), which, when intercalated into DNA, fluoresce under ultraviolet light (UVP Gel Documentation, Upland, CA, USA). Samples that contained *E. faecalis* DNA showed positive amplifications of 1522, 803, and 650 base pairs [34].

### 2.6. Statistical Analysis

Due to the uneven distribution of results, the Mann–Whitney U test was used. The Kruskal–Wallis test was used to compare the quality of the apical seal of MTA and BC-RRM, regarding the used technique, i.e., ultrasound or Er, Cr: YSGG laser). The IBM Statistics 19.0.0.1 software package was used (Statsoft, Tulsa, OK, USA) for the analysis. The level of significance was set at *p* < 0.05.

## 3. Results

After the incubation period, all samples from the positive control group showed viable bacteria in the root canals. There were no viable bacteria in the negative controls.

Table 1 showed the number of *E. faecalis* CFUs (minimal, maximum, and median) in root canals in all subgroups and the number of positive samples (samples with viable bacteria in root canal) in each group.

The BC-RRM and the MTA filled in ultrasonic retro cavities showed similar results in the growth of CFUs (*p* > 0.05). Similar results were found (CFUs) in the Er,Cr:YSGG group filled with the MTA (*p* > 0.05). The BC-RRM filled in Er, Cr:YSGG laserretro cavities showed the highest number of CFUs, suggesting a higher leakage (*p* < 0.001).

All positive results obtained by the culture method (growth of *E. faecalis*) were confirmed by PCR showing the presence of *E. faeaclis* DNA in all positive samples. In the Er,Cr: YSGG/MTA group, the DNA of *E. faecalis* was not detected in three samples. In the US/MTA group, six samples did not have *E. faecalis*, according to PCR analysis. In the US/BC-RRM group, all samples had present *E. faecalis*, although the culture method showed the growth in only five samples. In the Er,Cr: YSGG/BC-RRM group, the PCR analysis confirmed the results of the culture method, with all samples being positive.

## 4. Discussion

The aim of this study was to compare the bacterial leakage of two bioactive root-end filling materials in retro cavities prepared ultrasonically or with Er:YAG laser. The leaking model was first described by Antunes et al. [29] and is based on the assessment of the supply of the inoculated bacteria in the inner chamber (root canal) by nutrients from the outer chamber (Eppendorf tube), through the apical channels in the evaluated material. As explained in the study by Antunes et al. [29], since there are no nutrients available in the study, quality apical sealing prevents the seepage of serum to the canal, as well as the supply of bacteria, thus decreasing their number during the incubation period. The effectiveness and validity of the model were also confirmed in our study, showing an increase in the number of viable bacteria in all positive controls. In the negative controls, all root canals were without viable bacteria counts, thus proving the bacteria can only leak through the main canal. Additionally, it is possible that *E. faecalis* in conditions without nutrients transformed to a low-starvation and non-cultivable state. Therefore, PCR analysis was used as an additional microbiological analysis after the cultivation method.

Our results showed a similar apical sealing of the MTA and BC-RRM in ultrasonic retro cavities, which is in agreement with the studies by Antunesa et al. and Nair et al. [29,34]. However, compared to the culture method, PCR analysis showed better results for MTA, with six samples negative on *E. faecalis*, and the BC-RRM in US cavities leaked in all samples. In another study by Hirschberg et al. [28], bioceramic retro fillings leaked more than MTA in ultrasonically prepared retro cavities. The methodology is different in the mentioned studies regarding sample preparation and leakage models (the bacterial leakage, fluid filtration, and dye leakage models), so the results can not be directly compared. In this study, the bacterial suspension was inoculated into the previously instrumented and cleaned root canals, compared to the study of Nair et al. [34], in which, the *E. faeaclis* suspension was inoculated in root canals that had been retreated. However, the higher bacterial growth in the group filled with BC-RRM could be explained by the lower pH of the material, compared to MTA, which could be the possible reason for less growth of the bacteria in retrocavities filled with MTA [35].

In this study, the Er,Cr:YSGG laser preparation of retrovacities did not affect the sealing ability of the MTA; however, caused more leakage through BC-RRM material. Many previous studies have shown the advantages of erbium lasers in periapical surgery, in terms of increased apical sealing [6,12,36]. Karlović et al. [6] reported less apical leakage of MTA, Super EBA, and IRM materials in retro cavities prepared by Er:YAG laser, compared to ultrasonic. In a study by Kocak et al. [10], retro cavities prepared by the Er,Cr:YSGG laser leaked less, compared to cavities prepared ultrasonically. These conflicting results can be justified by the different leakage models used in the studies, i.e., the dye and liquid filtration methods. The advantage of the dye leakage method is the low molecular weight of the dies, which can penetrate the place where bacteria can not [37,38], while, in the method of fluid filtration, changes in pressure or the time interval can affect the results [39]. Greater leakage of the BC-RRM in the Er,Cr:YSGG laser retro cavities could be the result of lower bound and interaction between high viscous BC-RRM and laser irradiated dentine tissue. The PCR analysis also confirmed higher leakage of BC-RRM when placed in the cavities prepared with the Er,Cr:YSGG laser. After irradiation of the dentine surface with erbium laser, it becomes irregular and covered by a thin layer of carbonization, which could have compromised the bound with BC-RRM [40,41]. This could be particularly evident when the Er;Cr:YSGG laser is used, since its irradiation penetrates three times deeper into the tissue and causes pronounced thermal changes, with a less efficient ablation effect [42]. The negative thermal effect of laser irradiation (Nd:YAG laser) on the sealing ability of MTA sealer has already been reported [43]. Additionally, compared to ultrasonic, the dentine surface after erbium laser ablation is smooth, without a smear layer and with open dentinal tubules [44].

Another possible explanation could be the lack of water in dentine, caused by the thermomechanical effect of erbium laser radiation, which is necessary for the setting and mechanical properties of bioceramic materials [45]. However, more studies are necessary to clarify these results and the effect of laser irradiation on the sealing ability of bioceramic materials at dentine.

The limitation of the study is related to the value of the leakage tests through filled root canals [46]. Namely, one of the potentially confounding factors is the possibility that the tested filling material has an antimicrobial effect against bacteria in the leakage test, thus impeding their leakage through the filled root canal [46]. Furthermore, it should be taken into account that the leakage path can be through the outer surface of the root and not the filled canal. Pathways of microbial leakage were not followed histologically in this study.

## 5. Conclusions

The present study showed that MTA and BC-RRM filled in ultrasonic root-end cavities and MTA in laser-prepared cavities showed similar sealing ability. The BC-RRM showed more leakage when used in Er,Cr:YSGG laser retro cavities.

## Figures and Tables

**Table 1 bioengineering-09-00314-t001:** Number of *E. faecalis* CFUs and the number of positive samples in each of the experimental subgroups (n = 10/each).

Groups	Positive Samples	Range	Percentiles		
			25th	50th (Median)	75th
US/BC-RRM	4	0–6 × 10^6^	0	0	2.8 × 10^5^
US/MTA	2	0–4 × 10^4^	0	0	1.0 × 10^3^
Er,Cr:YSGG/BC-RRM	10	1 × 10^3^–7 × 10^7^	9.25 × 10^4^	1.7 × 10^6^	4.5 × 10^6^
Er,Cr:YSGG/MTA	1	0–6 × 10^3^	0	0	0

US—ultrasonic preparation of root-end cavity. Er,Cr:YSGG—root-end cavities prepared with Er,Cr:YSGG laser. MTA—mineral triokside aggregate. BC-RRM—bioceramic based material for retrograde filling.

## Data Availability

Data sharing not applicable.

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
