# Peer review of "Sealing Ability of Bioactive Root-End Filling Materials in Retro Cavities Prepared with Er,Cr:YSGG Laser and Ultrasonic Techniques"

_bioengineering, 2022, doi:10.3390/bioengineering9070314_

Round 1

Reviewer 1 Report

In this manuscript, the authors report and compare the sealing ability of materials.

Dear Sir/Madam

Manuscript ID:  bioengineering-1721431

The following are the comments:

  1. Introduction section can include relevant literature on the methods involved to understand sealing ability of materials
  2. Materials and Methods section should include the clearance obtained from human ethical committee
  3. Results and discussion: PCR analysis data need to be included

Manuscript is advised for Minor revision

Author Response

Point 1: Introduction section can include relevant literature on the methods involved to understand sealing ability of materials

Response 1: Thank you for your comment. We added relevant literature on the methods evaluating the sealing ability of materials. (Page 2, Line 67)

Point 2: Materials and Methods section should include the clearance obtained from human ethical committee

Response 2: The Ethical Committee approval was added in the Methodology.

Point 3. Results and discussion: PCR analysis data need to be included

Response 3. Thank your for your comment. The information regarding PCR was added.

Reviewer 2 Report

1.- This work is very interesting because it compares 2 materials that are very used in retro-filling cavities. 1st and 2nd generation of bioceramic materials

  2.- Some failures are due to the material but others could be to how the cavity is prepared  

3.- The strength of this work is to compare a new technique (laser) for retro-preparation and compare it with the classic ultrasonic technique and compare 2 excellent materials  

4.- The weakness is the sample size and that the authors should talk more about the effect of this laser in dentin and how this could affect the adhesion between Bc-RRm and dentin

Author Response

Point 1. The weakness is the sample size and that the authors should talk more about the effect of this laser in dentin and how this could affect the adhesion between Bc-RRm and dentin

Response 1. Thank you for your comment. Some more explanation of the laser effect on dentin and its influence on the sealing ability of BC-RRM was added in discussion .

Reviewer 3 Report

Dear Authors,

the paper is very interesting and can be considered for publication in bioengineering. However, before acceptance, some minor revisions are needed. In particular:

1) Please discuss if such methods can be particularly useful in systemic patients, like oncological ones. Please cite PubMed ID26862696

2) Please discuss if an accurate oral hygiene protocol can improve the results in vivo. Cite PubMed ID28696070

3) Please emphasize the role of chlorexidine in the management of perioperative period and in the maintenance of an accurate oral hygiene. Please cite PMID: 23216882

Author Response

Point 1.  Please discuss if such methods can be particularly useful in systemic patients, like oncological ones. Please cite PubMed ID26862696

Thank you for your question. Microbiological leakage test can be used only in laboratory in vitro studies and according to our best knowledge, it can not be used in clinical analysis.

Point. 2. Please discuss if an accurate oral hygiene protocol can improve the results in vivo. Cite PubMed ID28696070

Thank you for your question. Since the tested materials are used only in retrograde cavities and its success depends on their sealing ability, the improved hygiene protocol could not influence or improve their mechanical properties. However, the success of endodontic surgery procedures depend on the infection control during previous root canal treatment and endodontic surgery protocol Since this was an in vitro study, we did not evaluate the benefit of improved hygine protocol on the success of endodontic surgery.  

Point 3. Please emphasize the role of chlorexidine in the management of perioperative period and in the maintenance of an accurate oral hygiene. Please cite PMID: 23216882

Thank you for your question. Since this was and in vitro study, we did not evaluate the influence of chlorhexidine application on the success of the endodontic surgery protocol. This was an in vitro study in which the sealing ability of the new bioceramic material was tested.

Reviewer 4 Report

The aim of the present article was to evaluate the sealing ability of two different root-end filling materials in retro-cavities prepared by means ultrasonic or laser techniques.

I suggest corrections and comments:

-       Please check authors departmental affiliations 

Abstract

-       Authors should improve the abstract, specifying the statistical tests used in the study and reporting better the obtained results. 

-       Line 23: the description of the acronym PCR is missing

-       Line 25: check the error “(p   0.001)”.

Introduction

-       Reference 5 is missing along the text.

-       Line 51: Explain the composition of the MTA to make it more comprehensible to readers.

-       Please check the acronyms along the text.

-   In this section I suggest adding a short introduction on the analytical tools generally used in the comparison of these materials/techniques.

Materials and Methods

-       Line 79 and Line 150: authors should present the cited protocol in the text “[29]”.

-       Lines 79-81: “The root canals were instrumented…until 80 instrument X3 (30/07)” I suggest describing in detail all the used rotary instruments in sequence "until instruments X3".

-       Lines 104-106: Author should rephrase this sentence “In the positive….described before”.

-       Line 114: “root canals” check the plural; in this point should add the sample size of subgroups (n=10?)

-       Line 133: "the control of sterilization control ...samples" authors should explain the concept better

Results

-       Line 174: please replace the comma with the period in “(p<0,05)”

-       Line 182-185: authors should improve the results presentation.

Discussion

-       What are the limits of the study?

References

Check and standardize the entire bibliography according to the guidelines of the journal.

Author Response

Response to Reviewer 4 Comments

Abstract

-       Authors should improve the abstract, specifying the statistical tests used in the study and reporting better the obtained results.

Thank you for the comment. It has been improved according to your recommendation.

-       Line 23: the description of the acronym PCR is missing

It was added.

-       Line 25: check the error “(p   0.001)”.

It was checked and corrected.

Introduction

-       Reference 5 is missing along the text.

Thank you for your comment. The reference 5 was added in the text.

-       Line 51: Explain the composition of the MTA to make it more comprehensible to readers.

Thank your for your suggestion. The composition of MTA was added in the text.

-       Please check the acronyms along the text.

Thank you for your recommendation. They were checked and corrected where needed.

-   In this section I suggest adding a short introduction on the analytical tools generally used in the comparison of these materials/techniques.

Thank your for your suggection. The small paragraph was added before the aim of the study.

Materials and Methods

-       Line 79 and Line 150: authors should present the cited protocol in the text “[29]”.

Thank your for the suggestion. The protocols were added.

-       Lines 79-81: “The root canals were instrumented…until 80 instrument X3 (30/07)” I suggest describing in detail all the used rotary instruments in sequence "until instruments X3".

Thank your for your suggestion. We explained all the used instruments

-     Lines 104-106: Author should rephrase this sentence “In the positive….described before”.

The sentence was rephrased.

-      Line 114: “root canals” check the plural; in this point should add the sample size of subgroups (n=10?)

Thank you for your comment. It was corrected.

-      Line 133: "the control of sterilization control ...samples" authors should explain the concept better

Thank your for your comment. The streilization control was made using the culture method in order to confirm the sterilization of all samples before the experimental microbiological procedures.

Results

-       Line 174: please replace the comma with the period in “(p<0,05)”

It was corrected.

-       Line 182-185: authors should improve the results presentation.

Thank your for your comment. The paragraph was improved.

Discussion

-      What are the limits of the study?

Thank you for your comment. The limitations of the study were added in the last paragraph in the manuscript. The limitation of the study is the related  the  recently discussed value of laboratory  studies  dealing  with  leakage  through  filled root  canals Rechenberg et al. Int Endod J. 2011;44:183-94). A potential  confounding  factor  is  the  possibility  that  tested root filling  materials  could   impede   microbial  meakage  because  of their antimicrobial  properties and not leakage (ref). Another factor that should be taken into consideration is the route of leakage that besides the filled root canal could be the outer root surface. The   routes   of microbial    leakage    were    not traced    histologically in this study.

Rechenberg DK, De-Deus G, Zehnder M. Potential systematic error in laboratory experiments on microbial leakage through filled root canals: review of published articles. Int Endod J. 2011 Mar;44(3):183-94.

References

Check and standardize the entire bibliography according to the guidelines of the journal.

The references were checked.